# Position: Meaning Is Not A Metric: Using LLMs to make cultural context legible at scale

## Abstract

This position paper argues that large language models (LLMs) can make cultural context, and therefore human meaning, legible at an unprecedented scale in AI-based sociotechnical systems. We argue that such systems have previously been unable to represent human meaning because they rely on thin descriptions: numerical representations that enforce standardization and therefore strip human activity of the cultural context that gives it meaning. By contrast, scholars in the humanities and qualitative social sciences have developed frameworks for representing meaning through thick description: verbal representations that accommodate heterogeneity and retain contextual information needed to represent human meaning. While these methods can effectively codify meaning, they are difficult to deploy at scale. However, the verbal capabilities of LLMs now provide a means of (at least partially) automating the generation and processing of thick descriptions, potentially overcoming this bottleneck. We argue that the problem of rendering human meaning legible is not just about selecting better metrics, but about developing new representational formats (based on thick description). We frame this as a crucial direction for the application of generative AI and identify five key challenges: preserving context, maintaining interpretive pluralism, integrating perspectives based on lived experience and critical distance, distinguishing qualitative content from quantitative magnitude, and acknowledging meaning as dynamic rather than static. Furthermore, we suggest that thick description has the potential to serve as a unifying framework to address a number of emerging concerns about the difficulties of representing culture in (or using) LLMs. In addressing these challenges, we present a pathway to developing systems that can better represent and support meaningful human experiences across domains including healthcare, education, and sustainability.

# 1  Introduction

The premise of this paper is that it is important, but difficult, to develop technologies that can support meaningful human experience. Our position aims to articulate how AI technologies can be used not just to yield superficial increases in convenience, but to offer something deeper: a tool for helping people tap more directly into *what really matters*. This is not an inevitable outcome of technological progress. Dystopian fiction is full of cases in which a hypothetical society has advanced far beyond our own, only to find that their sophisticated technology has had a detrimental effect on human experience. Of course, the easy part is proposing that technology should have positive societal impacts; the difficult part is defining "what really matters."

For example, consider the case of social media. Initially these platforms were presented as a tool for human social connection [10]. Over time, however, they came to reflect the more superficial dynamics of engagement metrics; content was prioritized not to facilitate more profound social interactions, but on the basis of likes, views, and shares [103, 30, 63]. This led to a wide range of negative societal outcomes, such as rampant misinformation [55, 106], increased feelings of loneliness [78, 101], addictive behavior [40], and other psychological and social costs [22].

In response, technologists, social scientists, and activists have proposed other methods for prioritizing content on these platforms—ones that would, in one form another, allow us to more directly target what really matters [34, 97, 19]. A representative example is the Time Well Spent movement (now the Center for Humane Technology) led by Tristan Harris [34]. The goal of this movement was to develop technology that supports "meaningful" social connections and "choices we cherish" [72]. In response, at least one major social media platform alleged to have changed their recommendation algorithm to prioritize these kind of "meaningful" interactions [96]. However, it was disputed whether this effort had the intended effect, or simply substituted one superficial metric for another [72, 63].

While laudable in theory, proposals about orienting technology around meaningful experience have remained difficult to implement in practice. This is not a new problem, nor is it limited to social media platforms. For instance, in the early twentieth century Bertrand Russell predicted that technological development would soon lead to the widespread adoption of a four-day work week, thus allowing people to spend less time on drudgery and more time on what they actually care about [84]. Despite continued technological advancement, Russell's prediction has yet to materialize.

We contend that in order to develop sociotechnical systems that can increase time well spent or reduce drudgery, we first need a way of systematically codifying what kind of experiences humans find meaningful. We approach this problem—how to design sociotechnical systems that support human meaning—as one of representation. The core of our argument is that this issue has been difficult to address not simply because we have been using the wrong metrics, but because "metrics" (as we have traditionally conceived of them) are the wrong representational format. The formats we typically use for measuring behavior at scale exclude key contextual information necessary for representing human meaning. The primary aim of this paper will be to describe why that is the case and how we can begin to address it.

**Our position is that we can use LLMs to make human meaning legible at an unprecedented scale; that "thin" metrics alone will not be able to do this, because they cannot adequately represent cultural context; and that developing "thick" representational formats might eventually allow our society's most powerful and impactful sociotechnical systems to support the heterogeneous range of experiences that really matter to people.** We argue that such systems have previously been unable to represent human meaning because they typically rely on thin descriptions: numerical representations that enforce standardization, and therefore strip human activity of the broader context that gives it meaning [9, 71, 59, 89]). By contrast, scholars in the humanities and qualitative social sciences have developed robust frameworks for representing meaning based on thick descriptions: verbal representations that allow for heterogeneity, and therefore retain crucial contextual information necessary for analyzing cultural and experiential meaning [27, 108, 87, 82]. These methods provide a way of codifying meaning—but they are difficult to deploy at scale.

Previously, reliance on thick descriptions would lead to a bottleneck: only human analysts had the necessary verbal capabilities to generate and evaluate such representations. With LLMs, this is no longer the case. They provide a means of (at least partially) automating the generation and evaluation of thick descriptions. Overall, we make the case that meaning is not a metric: thicker representational formats are needed to render human meaning legible at scale.

## 2 The Legibility of Meaning

In his classic political science text *Seeing like a State*, James C. Scott (1998) introduces the concept of legibility [89]. This term describes the connection between a sociotechnical system (in Scott's case, a political state) and its information processing capacities. Political states cannot represent the goals and desires of their constituents directly. Simply asking each person what they want would result in too much heterogeneity of response. Rather, the state must develop standardized metrics that can be applied across the whole population. These help the state understand the overall trends and dynamics of the populace. However, as Scott argues, by enforcing this standardization the state makes some kinds of information legible—while rendering other kinds illegible.

Drawing on Scott's concept, we frame the main challenge of this paper as how to make human meaning *legible* at scale. To clarify: we are not concerned with how to make human meaning legible within AI models, but rather how AI models can be used to render new kinds of information legible within sociotechnical systems. We use the term *sociotechnical system* to refer to any large-scale configuration of people and machines that cannot be understood in terms of human or technological dynamics alone but that depend on a complex entanglement of their interactions [41, 5, 100]. This includes many systems that have a major impact on society, such as political states, social media platforms, corporations, and institutions for healthcare, education, and science. As AI technologies become increasingly integrated into these societally impactful systems, how can we use these technologies to make new kinds of information (such as human meaning) legible at scale?

### 2.1 What Do We Mean By "Meaning"?

The term "meaning" has been used by many different people, across many disciplines, to refer to many different (though often overlapping) concepts [27, 11, 4, 33, 23, 36, 24]. While there is undoubtedly heterogeneity in these different usages, there are also recurring themes—such as the relational, contextual, and contested nature of meaning. In this paper, we are primarily concerned with what we call "human meaning," to distinguish our usage from debates about whether LLMs really "understand" the *semantic* meaning of the text they generate [6, 50]. We take human meaning to have two components: cultural and experiential.

*Cultural meaning.* This is the endorsement of certain symbols as significant within the context of a given community. These symbols could include cultural emblems (such as the Union Jack or the Olympic rings), behaviors (such as bowing or raising one's middle finger), rituals (such as a marriage ceremony or graduation procession), events (such as the Battle of Waterloo or the Moon landing), or works of art (such as James Joyce's *Ulysses* or Picasso's *Guernica*). This is how the term is typically used in humanities and social science fields such as literary studies [32], history [17], and cultural anthropology [27]. This usage is consistent with other usages of "culture" in contemporary AI [21, 113, 86, 1, 44]. For our purposes, this is the kind of meaning we can most directly make legible with existing frameworks from the humanities and social sciences.

*Experiential meaning.* This is the individual mental state of believing a particular event or experience to be especially worthwhile or to matter in some grander sense. It is how the term is typically used in fields such as psychology and mental health [36, 105]. More informally, this is the sense used by religion, self-help, and other pathways for orienting one's life toward *what really matters* [24]. Often, experiential meaning is framed as a magnitude estimate: whether a given experience is more or less meaningful than another one [38, 37, 94]. However, these magnitude judgments often miss key aspects of the cultural basis of meaning [11, 12, 14, 15, 35, 104].

It is not always easy to disambiguate between cultural and experiential meaning. For example, experiential meaning derives from cultural meaning in ways that are crucial but often difficult to systematize [13, 52, 114]. The more we understand about a person's cultural background, the more likely we are to understand what they find personally meaningful. But while people's individual sense of meaning is shaped by their cultural milieu, their individual perceptions of meaning can either conform to a given cultural template—or reject it. While the framework we describe below can be used to codify experiential meaning [11, 60, 109, 82], for the purposes of this paper we will focus primarily on cultural (rather than experiential) meaning, since that is traditionally where thick description has been most directly applied. We see this as a crucial first step toward making human meaning, in its broadest sense, legible within sociotechnical systems.

## 3  Representing Meaning via Thick Description

In this paper, our primary reference point for meaning is Clifford Geertz's (1973) account of "thick description [27], which provides a framework for the interpretation of symbols and behaviors within their cultural context to uncover their layers of significance. Methodologically, it is a bedrock of qualitative approaches in the social sciences [87, 64, 58]; theoretically, it is a unifying framework for influential schools of thought in disciplines such as literary criticism [32], history [17], political science [108], philosophy [98], and more. It is arguably one of the most influential frameworks in the humanities and social sciences.[1] Thick description offers a conceptual and methodological bridge between cultural and experiential meaning, as well as a guide for how human meaning can be rendered legible—both to human interpreters and computational systems—at scale.

Geertz proposed thick description as a means of clarifying the goal of ethnographic fieldwork in cultural anthropology. The aim is not just to provide a surface-level account of physical behaviors, rituals, stories, or cultural norms (i.e., *thin* description). Rather, it is to provide an interpretation of the symbolic dimensions of a given behavior or practice—to analyze its role in providing an individual with a larger sense of meaning and purpose in the context of their broader community (i.e., *thick* description). Geertz contended that this approach to the study of culture is "not an experimental science in search of law but an interpretive one in search of meaning" [27].

An example of the contrast between thin and thick description can be seen in different ways of describing an economic activity. For instance, an entire country's economic activity can be encapsulated in the useful, if incomplete, measure of gross domestic product. This thin description can be used to track whether a nation's economy is trending in a positive or negative direction but does not provide the language to distinguish between, for example, the economic considerations of purchasing a house (a major outlay of monetary resources) versus the experiential considerations of a buying a home (a significant choice about the composition of one's family, lifestyle, and community).

By contrast, a canonical instance of thick description is Malinowski's (1922) investigation of the Kula Ring [57, 29]. The Kula Ring refers to an elaborate ritual in which inhabitants of the Trobriand Islands crossed treacherous swathes of water in small boats to trade necklaces and armbands which in themselves had no obvious utility: the necklaces and armbands were never actually worn. Malinowski's account showed how this ostensibly economic activity could only be understood by an analysis of its symbolic dimensions. That is, it cannot be accounted for by an analysis of the transactions through an externally observable value metric (such as money). Rather, it requires an analysis of the meaning assigned to such transactions internally by the people participating in them. The Kula necklaces and armbands were symbols of achievement and social standing. Comparably, an observer attempting to understand the value of a Super Bowl trophy in terms of how much water or soup it holds will likely miss important aspects of the broader cultural activity it represents.

As a theory of meaning, thick description operationalizes meaning as a relationship mediating a symbol (including artifacts such as a Super Bowl trophy or the Bible; actions such as attending a protest or making a particular hand gesture; and mental states such as belief in the sanctity of marriage or satisfaction with clearing one's email inbox) and its sociocultural context. Influenced by Wittgenstein (1953), Geertz claimed that meaning is not an intrinsic property of an individual mind but only exists within the broader context of who is interpreting a given symbol, the background or experience they bring, and the time and place in which the interpretation occurs [27, 110].

As a result, the meaning of any symbol is inherently multiplicitous: no singular meaning prevails, but rather multiple meanings can coexist depending on the interpreter and their sociocultural conditions. For example, a hand signal such as a "thumbs up" may signify approval in Western cultures, but in parts of the Middle East and Africa can be perceived as an offensive expression of contempt. The meaning of a symbolic figure can, therefore, only be understood against its appropriate cultural ground: what makes a description "thick" is the extent to which it provides the necessary contextual grounding to locate a particular meaning.

---

[1]According to Google Scholar's citation count [20 May 2025], Geertz (1973) has >100k citations. For comparison, foundational frameworks in related disciplines published around the same time include: Kahneman & Tversky's (1979) prospect theory in the behavioral sciences (89k citations) and Granovetter's (1973) weak ties social network analysis in the quantitative social science (76k citations).

## 3.1 Thick and Thin Description in Sociotechnical Systems

Thus, one promising conceptual framework for making meaning legible at scale is to make thick descriptions legible at scale. However, the default format of representation within most sociotechnical systems is thin; generally speaking, only numerical signals—canonical thin descriptions such as GDP, engagement metrics, or stock price—are legible within the information processing capacities of such systems [9, 71, 59, 89]. In this section, we argue that, across a wide range of sociotechnical systems, the use of thin descriptions strips away the kind of contextual data needed to encode a signal of human meaning.

The distinction between thin and thick descriptions is related, but not identical, to the distinction between quantitative and qualitative data. Thin descriptions account for facts, figures, or observable behavior without context; thick descriptions account for the context, interpretations, subjective perspectives, and meanings that make sense of those facts. Not all qualitative data conveys a thick description; otherwise, it would not be necessary to have a theory of what makes a given qualitative analysis more or less thick. Conversely, while thick description traditionally refers to qualitative analysis, it is nonetheless possible for a quantitative figure to capture thick elements (e.g., with multiple variables). The distinction is not binary: a representation can be "thinner" or "thicker."

The key problem is that, historically, the only way to reach legibility at scale is via thin description. Standardization must be applied, such that every individual performance can be assessed in the same generic terms. However, this kind of standardization is antithetical to encoding meaning via thick description. If concepts can only be made legible within a system on the condition of standardization, then that creates a natural tension with the context-rich information necessary for a thick description.

For example, it is often the case that the primary goal of a system is qualitative: to promote health, education, urban livability, social connection, justice, and so on. But what is legible within the system is quantitative (e.g., body mass index, standardized test scores, crime statistics, social media shares, or recidivism rates); this quantitative metric becomes a tangible proxy for the more abstract, multi-faceted goal. While such data-driven optimization is a central aspect to the functioning of modern sociotechnical systems [75, 71], there are well-documented issues with overreliance on such quantitative signals [66, 43]. We argue that relaxing this mandate for quantitative standardization by creating representations based on thick description can help overcome these issues.

Healthcare is a prime example. Many conceptual definitions of health take a thick perspective. For example, the Constitution of the World Health Organization defines health as "a state of complete physical, mental and social well-being and not merely the absence of disease or infirmity" [111]. Likewise, several prominent theories of health and well-being emphasize the functional nature of health—that what constitutes healthiness depends on what is necessary to physically support a meaningful life [99, 90, 70]. This definition of health is highly context-dependent: there is no single standardized template for health, just as there is no standardized template for the life people want to live. However, hospitals and healthcare systems need tangible metrics by which to measure their efficacy, such as length of stay, readmission rates, and morality rates. Though important, over-reliance on these thin metrics runs the risk of missing crucial qualitative aspects of care such as emotional, familial, occupational, and social facets of human life [74]. Thicker representations are needed to codify this context-dependent, holistic notion of health in a large-scale sociotechnical system [73].

This basic pattern holds across a range of domains. In education: standardized test scores, grade point average, and graduation rates fail to capture student growth, creativity, critical thinking, and character development [80, 51]. In scientific research: journal impact factors and citation counts attempt to quantify research impact but can miss crucial aspects like conceptual innovation, methodological rigor, societal impact, the authors' capacity for mentorship, and influence on society outside traditional citation patterns [77, 20]. In sustainability: a narrow focus on carbon emissions metrics (though important) can miss deeper notions of sustainability based on cultural and spiritual connections to land, traditional ecological knowledge, and community-specific environmental practices [47]. In employee hiring and evaluation: quantitative metrics like sales numbers, tasks completed, or hours worked fail to capture crucial qualities like mentorship, team morale building, institutional knowledge sharing, and maintaining positive client relationships [76].

In each of these cases, there is a more abstract notion of "what really matters" that gets consolidated into a thin signal which is legible at scale. We contend that employing thick description can help build systems that can represent something closer to the meaningful target notion.

# 4 Scaling Thick Description: Five Challenges for Measuring Meaning

In the previous section, we argued that robust strategies for representing meaning already exist. They are used by scholars in the humanities and qualitative social sciences [27, 87, 82]. Accordingly, we contend that if it were possible to employ a sufficiently large number of literary critics, cultural anthropologists, historians, or other relevant scholars within large-scale sociotechnical systems, then these systems would have robust capabilities for representing and processing information related to cultural context: in other words, to thickly describe human meaning at scale.

Our proposal is to use LLMs to fulfill at least part of the functionality of human experts in these domains—to act as cultural critics at scale. Working in conjunction with human experts, LLMs can help solve the problem of scaling thick description. By prompting LLMs to produce thick descriptions, this approach would seek to reproduce a form of analysis that has typically required deep, situated human judgment. Crucially, this strategy operates within existing computational architectures, without requiring a fundamental redesign of current AI technologies. However, the feasibility of this approach depends on addressing two key challenges: first, the conditions under which LLMs are empirically capable of producing thick descriptions that approximate human-level interpretations; and second, whether such descriptions can be systematically translated into forms that are actionable within computational systems without erasing their cultural and interpretive complexity. We articulate how progress could be made on this front by describing five key challenges for measuring meaning, and for each one offer initial ideas for paradigms that would address them.

## 4.1 Meaning Only Exists in Context

The thrust of our proposal is not that qualitative representations are better than quantitative ones. It is that in order to represent meaning, we need to represent context. There are many ways to do this, with qualitative thick description being a canonical one. But for any given implementation strategy, the standard is not whether it is qualitative or quantitative—but how well it retains information about context in which the meaningful experience or action originally occurred.

This context-dependence of meaning holds across a range of domains and definitions of "meaning." For example, words are meaningful, but only when used in the context of a given language; a swastika is a meaningful symbol, but its meaning changes when found in Western Europe versus a Buddhist country; and marriage ceremonies can be meaningful, but what specific aspects of the ceremony will be perceived as meaningful depends on the culture in which it is taking place—not to mention the individuals getting married. The difficulty is heterogeneity. What kind of context needs to be represented in order to capture meaning is underconstrained. Representing context is not simply a matter of listing proxy variables, like country, language, or ethnicity, nor can it be encapsulated in a feature set of norms, preferences, or taboos [113, 44, 85]. This is why standardization is antithetical to the representation of human meaning.

One means of translating between thinner and thicker representations is via novel methods for qualitative coding, in which subjective interpretations are condensed into a numerical value by a human analyst [87]. Indeed, the question of translating between thinner and thicker descriptions can be reframed in terms of such coding: How do we create domain-general methods for coding qualitative contextual factors into numerical values? Traditionally, qualitative coding schemes are bespoke for a given dataset—a human researcher must develop, test, and train other researchers to use them. However, with LLMs it may be possible both to develop a unique coding scheme for relevant aspects of a given context and then to employ the newly created scheme [49, 88]. This could help avoid the mandate for standardization by allowing for an ad hoc process of evaluation, adaptable to an arbitrarily large number of contexts, which nonetheless yields numerical values.

## 4.2 There Is No Single Source Of Truth for Meaning

The point of codifying meaning is not that there are "right" and "wrong" answers; it is that there are no *easy* answers and that any oversimplification necessarily omits crucial information about human meaning. The same symbol, artifact, or experience can carry different meanings depending on who is interpreting it and in what context. Systems for representing meaning should maintain this plurality rather than collapsing diverse interpretations into singular representations.

For example, consider the case of a student for whom we have comprehensive educational data, including standardized test scores, demographic information, personal essays, psychological evaluations by third-party experts (think: an augmented college admissions application). One evaluator might interpret an application as providing evidence of technical focus and emerging leadership in STEM fields. Another might view it as the story of a creative, emotionally attuned individual with potential in design or writing. A third might focus on signs of perfectionism or risk-aversion. Each interpretation is grounded in the same underlying data, yet each foregrounds different elements of meaning. This highlights a core difficulty: even when the input is constant, thick descriptions can diverge depending on the interpretive lens. The goal, then, is not to eliminate this divergence but to build systems that can recognize and represent such pluralism. How do we apply interpretive judgment at scale in ways that are consistent, fair, and reflexive?

Recent work shows how LLMs can be used to reckon with this kind of ambiguity, specifically to identify the presence of dog whistles [61, 53]. These are a form of coded communication that uses covert social or ideological signals to convey a generally acceptable meaning to a wider audience while conveying a more controversial or offensive meaning to a targeted audience. This leads to conflicting, audience-dependent layers of meaning. This work shows that LLMs are capable of word-sense disambiguation by analyzing contextual usage, distinguishing between benign and coded meanings, giving a preliminary instance of how LLMs can be used to support interpretive pluralism.

## 4.3 Both Lived Experience and Critical Distance Are Crucial

Expert judgments from trained humanists and social scientists are useful, but they must be grounded in the lived experience of the individuals they are trying to understand. Geertz (1983) distinguishes between the concepts of *experience-near* (informal usages that reflect a subjective perspective) and *experience-distant* (formal usages that reflect an analytical or scientific perspective) [28]. For example, a gambler and an AI researcher might describe the same situation in different terms (e.g., experience-near: slot machine, hitting it big; experience-distant: n-armed bandit, sparse rewards). Both specialized scholarly knowledge (experience-distant) and lived cultural experience (experience-near) are necessary for robust representations of meaning. Neither should be privileged at the expense of the other, and systems should facilitate integration across these different forms of expertise.

A range of recent work could support this effort. In natural language processing, researchers have begun to explore perspectivist approaches: data sets that are labeled to represent the judgments of annotators with differing, sometimes conflicting, points of view—rather than labeling data according to a single ground truth [25, 2, 112, 93]. Similarly, work in pluralistic alignment has sought to create AI systems that reflect diverse values and perspectives [92, 91, 56]. Efforts like these could specifically target a balance of experience-near and experience-distant perspectives.

## 4.4 What ≠ How Much

A key problem in mapping between cultural and experiential meaning is the difference between content and magnitude. So far we have discussed qualitative methods for assessing meaning [27, 87, 82]. However, psychological studies of meaning typically use quantitative assessments: surveys in which respondents are asked to rate on a numerical scale how meaningful they perceive a given experience, activity, or event to be [94, 36, 38, 37]. These result in fundamentally different questions about meaning. Qualitative approaches ask a question about content: "what is it?" By contrast, quantitative approaches ask a question about magnitude: "how much is there?" Effective systems for representing meaning should maintain this distinction rather than reducing one aspect to the other.

At first glance, creating a mapping between these two facets of meaning may seem like a straightforward empirical problem; however, this is not the case [15, 35]. For instance, it might seem that the solution would simply be to gather enough human-generated data in which people rate experiences as more or less meaningful. With enough features coded into the data, it could be determined which experiences people find meaningful—and those could be supported through technological interventions. However, this kind of approach would have difficulty accounting for the ambiguity in asking participants about meaning versus happiness [69], the radical flexibility of the interpretive process underlying judgments of meaning [52], the difficulties of codifying context as a feature set [113, 44, 85], that the same events can be perceived as more or less meaning depending on their narrative framing [83], the discrepancies between self-report and revealed preference [63], among

many other difficulties in combining cultural and psychological perspectives on meaning [11, 98]. It therefore remains an open question how best to develop mappings from qualitative interpretations about the cultural content of meaning to quantitative estimates of its psychological magnitude.

### 4.5 Meaning Is Made, Not Found

Humanists and social scientists often talk about "meaning-making," a term which better captures the dynamic, dialogic process underlying judgments of meaning [26, 3]. Culture and meaning are not static; they evolve continuously through social negotiation and changing contexts [7]. Meaning-making may be usefully framed as a search process [24], but it is one through a space that is undergoing constant reorganization. What is meaningful can quickly become meaningless (or vice versa) as interpretive frames shift. Systems for representing meaning must be capable of adapting to these constant changes rather than treating cultural meaning as fixed.

Recent psychological evidence shows that people's judgments about meaning can change based on the narrative framing of their experiences [83]: the same events elicit different ratings of meaningfulness when they occur in the context of different stories. This suggests that not only will LLMs be useful for measuring meaning—but they will become participants in the meaning-making process. For example, they may influence people's perceptions about the meaningfulness of their own experiences through guided journaling [68, 46], personalized personas [18], or psychotherapy [67, 45]. Furthermore, how LLMs categorize people as having a shared contextual frame may have crucial downstream effects on their interactions with others [16]. Overall, this is consistent with what humanists and social scientists call the "reflexivity" of meaning: that the way we measure, codify, or understand meaning has a direct impact on the meaning-making process itself [31, 110, 8].

## 5 Discussion

In this paper, we have argued that LLMs can be used to make human meaning legible at an unprecedented scale. We contend that the primary obstacle to this effort is that human meaning can only be encoded via thick description—but, historically, the only way to reach legibility at scale is via thin description. We propose that LLMs can be used to make thick descriptions a useful representational strategy in large-scale sociotechnical systems. A crux of this is about adequately representing cultural context. Accordingly, there are useful gradations of thickness between descriptions that are fully thick and fully thin—and therefore many ways to usefully make progress on this problem. These gradations exist through relaxation of the mandate for standardization, mainly by finding ways to accommodate the inherent heterogeneity of socially-situated behavior.

### 5.1 Better Metrics Are Not Enough

Our proposal shares a premise with other calls to use AI in reshaping our society's sociotechnical infrastructure (such as recommender systems) to reflect a broader sense of human well-being [97, 95, 42]. However, these efforts are primarily concerned with identifying which proxy metrics will most effectively capture human well-being. Our position is that all metrics will inevitably succumb to proxy failure [43] and what is needed instead is an entirely new representational format (i.e., thick description). To illustrate, this is why scholars who are concerned with meaning in the humanities and social sciences write papers full of long verbal exposition rather than ledgers of numbers [27, 28, 29]. Our proposal is similarly aligned to the nascent field of representation engineering [115], which places the representation of information (rather than its processing) at the core of technical development.

### 5.2 Without Thick Description, We Cannot Represent Key Aspects of Culture

Several recent papers make compelling cases about the difficulties of representing culture in (or using) generative AI models [113, 50, 1, 44, 86, 54, 102, 21]. While most of these do not focus primarily on "meaning" in the same way that we do, they reach broadly similar conclusions about the difficulties of representing cultural context in a nuanced way. Our proposal—to use thick description as a framework for making human meaning legible at scale—offers a potential unifying framework for these various perspectives. Thick description is a representational strategy designed to accommodate heterogeneity. A commonality among these accounts is that generative AI models depend on [21, 50], fail to account for [1, 44, 112, 86], or are not sufficiently representative of [54, 102] this kind of

heterogeneous cultural information. We argue that thick description gives us a framework for making progress on this wide variety of issues in representing the nuance and diversity of human culture.

### 5.3 This Is A Crucial Application Of Approaches To Scaling Qualitative Methods

Social scientists have begun exploring a range of efforts to scale qualitative methods (including thick description) using LLMs and other forms of generative AI [79, 23, 88, 49, 65, 107, 81]. Some offer thick description (or other social science approaches) as a method for evaluating AI systems [79, 107, 88]. A notable case is Qadri and colleagues (2025), who argue that "thick evaluations" are necessary to measure whether AI can represent key aspects of minority cultures [79]. While our position is aligned with their work in many ways, we argue that thickness is needed to represent all cultures (not just minorities) and that it should be used as a representational scheme itself and not just for evaluation. Other efforts have used LLMs to conduct qualitative analysis on large corpora of culturally-situated text [23, 81, 49, 65]; building on this work will be foundational toward ensuring the reliability of deploying thick description at scale. Especially promising work in this vein uses qualitative methods in not just in data analysis but in the generation of data as well [62].

### 5.4 Alternative Views, Risks, and Points of Failure

The following views are not mutually exclusive with the position offered in this paper. Rather, they present crucial risks and points of failure in pursuing this line of research.

**This is going to lead to bad outcomes because you will fail at robustly representing meaning; like social media, you will start off trying to measure "what really matters" but ultimately end up with something much more narrow insidiously masquerading as the deeper thing.** As implied in our section presenting five challenges, a lot can go wrong when trying to represent meaning. It is easy to overextend a given generalization beyond where it would be useful, accurate, or beneficial. It is difficult to convey the essence of culture in a way that does not trivialize it [113, 44]. One way our proposal could go wrong is by claiming to liberate a sociotechnical system from "enforced standardization"—only to make such standardization harder to discern. The risk is overgeneralization: applying one cultural standard for meaning in a context (or to an individual) where it does not belong.

**This is going to lead to bad outcomes because you will succeed at robustly representing meaning; this will create a powerful lever not only for positive actors, but bad ones as well.** For example, if a political state has the power to use qualitative analysis of personal data for noble purpose, then it would also have the power to use it for nefarious purposes as well [48]. In *1984*, George Orwell imagined a future where "thoughtcrime" was enforced by legions of spies and informants. But the same thing could be accomplished if a large amount of personal data were available online and the state could employ the kind of automated qualitative analysis we describe.

**You claim to draw on humanities and social science frameworks. But thick description is only one such framework; others paint a much more complicated picture.** For example, Haraway (1988) argues that all knowledge is intrinsically partial, embodied, and dependent on specific historical and social contexts [33]. Thus, when the scientific community assumes that the perspective of its work is neutral and objective, it is simply imposing its own partial, contextually-bound perspective—usually a WEIRD one [39]—on an authoritative body of knowledge without having to acknowledge the imposition. According to Haraway, this paper's proposal could potentially provide a more powerful mechanism for imposing a singular way of seeing things on a diverse cultural ecosystem. Likewise, Barad (2007) argues that the act of making something legible within a sociotechnical system does not just observe it in a neutral sense, but in fact transforms it [4]. For example, when a recommender system measures user preferences, it is not just recording pre-existing preferences; it is creating the categories that make certain preferences possible, likely, or even inexpressible. According to Barad, this paper's proposal could potentially exert a corrupting influence on the original cultural phenomenon itself, pressuring it to fit into the categorical box that renders it legible within the sociotechnical system.

**This isn't a technical problem; it's a political one. Even if you could represent meaning in the way you're suggesting, there's no guarantee that powerful sociotechnical systems would employ such representations in a beneficial way. In fact, they probably wouldn't.** This is a significant concern. However, in order to achieve the political solutions we want, we first need to have the technological solutions to support them. Our position is offered in service of this effort.

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
