# OpenReview forum: "Position: Meaning Is Not A Metric: Using LLMs to make cultural context legible at scale"
_NeurIPS.cc/2025/Position_Paper_Track — Submitted to NeurIPS 2025 Position Paper Track_

### Official Review · Reviewer_5qWL · 2025-08-06

**Significance:** 4
**Presentation:** 4
**Rating:** 10
**Confidence:** 4

**Summary:**

The paper is concerned about making human meaning  legible at scale in sociothecnical sytems.

-Human meaning is defined in section 2.1 as made of 2 components: CULTURAL (symbols significant in a community, e.g. Olympics rings) and EXPERIENTIAL (experience of events that matter, e.g. coming to age ceremonies).

-Representation of Meaning is in section 3, it is based on Clifford Geertz’s classification of thin and thick descriptions. Thin description is intended as a quantitative representation that "strip[s] human activity of the cultural context that gives it meaning".

Thick description means  "verbal representations that accommodate heterogeneity and retain contextual information needed to represent human meaning."

-Legibility is in section 2, but I had to use Claude to get a meningful definition "[Scott's] legibility is the state's process of simplifying and standardizing complex social realities into formats that can be easily measured, monitored, and administered by bureaucratic institutions."

-Sociotechnical systems are states, social media platforms, institutions for healtcare, education, etc.

The positon of the paper is that LLMs can make thick descriptions legible at scale.

**Strengths:**

The paper provides a well presented description of several concepts related to the analysis of meaning.
In the thin/thick descriptions framework, authors explain their choice of representing Meaning via thick descriptions to fully, or, at least, better describe "what really matters" moving away from the tyranny of quantitative proxies (thin descriptions).

To make thick descriptions available/legible at scale would require the intervention of content-related scholars at levels not possible.

Their claim is that "LLMs can help solve the problem of scaling thick description. [...] this approach would seek to reproduce a form of analysis that has typically required deep, situated human judgment.", and, in this critical task of scaling thick descriptions, they identify (and discuss) 5 critical challenges for LLMs.
-Meaning Only Exists in Context
-There Is No Single Source Of Truth for Meaning
-Both Lived Experience and Critical Distance Are Crucial
-What $\neq$ How Much [Quantitative vs Qualitative]
-Meaning Is Made, Not Found

Contrasting views are discussed and addressed at a general level.

The paper provides an interesting and growing approach to AI models that entails deeper questions about cultural and social meaning.

**Weaknesses:**

As I had to navigate through concepts and definitions far from my technical background, I appreciated the clarity of many of those concepts in the paper, and the care, by the authors, to make readers like me comfortable in this environment.

The only concept that I found hard to grasp is legibility, and I would suggest the author to revisit that definition. Besides that I think the authors did a good job at addressing the core position.

There are no specific actions suggested in the paper, but this is consistent with the speculative  approach of the paper.

**Questions:**

I'd like to challenge the authors on how they would get started on a project like this. How would you design the implementation of a pilot project?

**Alternative Position:**

Yes, and alternative positions are well-considered and addressed by the argument

**Author Identification:**

No.

**Context:**

4

**Discussion:**

4

**Ethics:**

["NO or VERY MINOR ethics concerns only"]

**Position:**

Yes, the paper argues for or against a position related to machine learning.

**Support:**

3

**Thoroughness:**

5

---

### Official Review · Reviewer_fneG · 2025-08-10

**Significance:** 4
**Presentation:** 4
**Rating:** 8
**Confidence:** 4

**Summary:**

This position paper argues that LLMs can make human meaning legible at unprecedented scale by enabling "thick description" rather than relying on "thin" numerical metrics. Drawing on anthropological theory (Geertz) and political science (Scott), the authors contend that current sociotechnical systems strip away cultural context essential for representing human meaning through standardized metrics. They propose using LLMs to automate generation and processing of thick descriptions that preserve contextual information. The paper identifies five key challenges: preserving context, maintaining interpretive pluralism, integrating lived experience with critical distance, distinguishing qualitative content from quantitative magnitude, and acknowledging meaning as dynamic. Examples span healthcare, education, and sustainability domains.

**Strengths:**

The paper is exceptionally well-written and shows strong theoretical depth. It tackles a fundamental problem in representing human meaning in computational systems, offering a novel integration of ideas from anthropology and political science. Examples from multiple domains clearly illustrate the limitations of thin metrics. The five challenges are well defined and paired with thoughtful responses. Alternative perspectives and potential risks are addressed thoroughly. The interdisciplinary approach provides valuable insights for the ML community, and the position is clearly stated and persuasively argued.

**Weaknesses:**

- Lacks empirical examples or case studies demonstrating the practical utility of SMF in actual ML/NLP system design.
- The critique of computationalist approaches sometimes relies on strawman formulations, missing more nuanced defenses from recent literature.
- Limited engagement with cognitive science work that bridges symbolic/statistical models and embodied/situated perspectives.
- Practical implementation pathways for SMF are vague, leaving unclear how designers and engineers should operationalize these ideas.
- Evaluation criteria for “meaning alignment” remain abstract and not reproducible across domains.
- The argument is ambitious but risks alienating practitioners due to its heavy reliance on theoretical framing without actionable guidance.

**Questions:**

How would you empirically validate that LLMs can produce thick descriptions that accurately capture cultural meaning comparable to human experts? What would concrete pilot implementations look like across the domains you discuss, and what metrics would demonstrate their effectiveness without falling into the same traps as thin descriptions? How do you reconcile the need for interpretive pluralism with the practical requirements of actionable representations in computational systems?

**Alternative Position:**

Yes, and alternative positions are well-considered and addressed by the argument

**Author Identification:**

No.

**Context:**

4

**Details Of Ethics Concerns:**

The paper discusses representation of cultural context and meaning, with appropriate acknowledgment of risks including potential surveillance applications and cultural misrepresentation.

**Discussion:**

4

**Ethics:**

["NO or VERY MINOR ethics concerns only"]

**Position:**

Yes, the paper argues for or against a position related to machine learning.

**Support:**

3

**Thoroughness:**

4

---

### Official Review · Reviewer_hZRd · 2025-08-25

**Significance:** 1
**Presentation:** 3
**Rating:** 3
**Confidence:** 3

**Summary:**

The authors contrast thin and thick descriptions, and then argue that LLMs can be used to make thick descriptions legible at scale. Crucially, legible is a term of art -- it was introduced by Scott (1998) to initially describe the what kinds of information can be processed by a political state.

**Strengths:**

Clearly written, well explained. The paper introduces the concept of thick descriptions to an audience that is likely not familiar with thick vs thin descriptions. Several examples are given and key issues are raised.

**Weaknesses:**

The paper does not address the key issue raised by the "legibility of meaning", in the sense of legibility as a property of how information is aggregated within a state. Namely, for information to be legible, it must be aggregated -- hence why thin descriptions are commonly used.

The authors devote a whole section (section 4) to the difficulties in aggregating thick descriptions (meaning only exists in context, no single source of meaning, lived experience and critical distance are crucial, and what $\neq$ how much) but do not argue why or how LLMs are the appropriate tool to enable the aggregation of thick descriptions. Instead, the authors point out a few general uses for LLMs that are similar to the issues the authors rightly raise. For example, in 4.2 the authors state "Systems for representing meaning should maintain this plurality rather than collapsing diverse interpretations into singular representations" but only point towards recent work on using LLMs to identify dog whistles (a discriminative task) as a possible first step in supporting interpretative pluralism (an aggregation task).

**Questions:**

1. If the key problem with the legibility of thick descriptions is their aggregation so that they can be processed by a sociotechnical system (e.g. a political state), why do you believe that LLMs can aggregate thick descriptions?
2. Related to Question 1, throughout the paper LLMs are referenced, but not described. What description of an LLM leads you to believe that what is argued is possible? As the paper currently reads, LLMs are treated as magic, not a tool to be used to solve a problem raised.
3. More generally, chatbots and the LLMs that support them have been described as "mansplaining as a service". This is anathema to lived experience and critical distance. Why do you believe these tools could work for thick descriptions?

**Alternative Position:**

Yes, and alternative positions are well-considered and named but not addressed

**Author Identification:**

No.

**Context:**

4

**Discussion:**

3

**Ethics:**

["NO or VERY MINOR ethics concerns only"]

**Position:**

Yes, the paper argues for or against a position related to machine learning.

**Support:**

2

**Thoroughness:**

4

---

### Note · Authors · 2025-09-02

**1-10 Additional Comments:**

Citations from 1-9 Camera Ready Changes:

[1] Lee et al. (2024). Aligning to thousands of preferences via system message generalization. https://proceedings.neurips.cc/paper_files/paper/2024/file/86c9df30129f7663ad4d429b6f80d461-Paper-Conference.pdf
[2] De Paoli (2023). Writing user personas with Large Language Models: Testing phase 6 of a Thematic Analysis of semi-structured interviews. https://arxiv.org/abs/2305.18099
[3] Feng et al. (2025). SS-GEN: A Social Story Generation Framework with Large Language Models. https://arxiv.org/abs/2406.15695

**1-11 Submit Again:**

Definitely yes

**1-1 Submission Process:**

5

**1-2 Next Year:**

Could be interesting to select a couple position papers for a Behavioral and Brain Science style commentary format (with one target article and many short commentaries responding to it; see https://www.cambridge.org/core/journals/behavioral-and-brain-sciences/information/about-this-journal).

**1-3 Future Development:**

The emergency reviewer situation didn't seem ideal. Ours offered poor quality commentary. Perhaps it would be better to have two reviewers who had an opportunity to truly dig into a paper rather than find a third who doesn't go as deep.

**1-4 Interest:**

["Panel discussions with other position paper authors", "Structured debates on controversial topics"]

**1-5 Thoughtful:**

7

**1-6 Supportive:**

7

**1-7 Technical Aspects Versus Position:**

6

**1-8 Gate Keeping:**

10

**1-9 Camera Ready Changes:**

Two reviewers were very enthusiastic about our paper. One wrote that the paper "shows strong theoretical depth" and "tackles a fundamental problem in representing human meaning in computational systems." The other scored our paper as 10/10 "award quality" writing that we offer an "interesting and growing approach to AI models that entails deeper questions about cultural and social meaning." Both reviewers suggested minor adjustments (e.g., clarifying our definition of legibility) which we are happy to incorporate.

By contrast, the emergency reviewer was more critical. They offer a critique with a singular focus on whether or not LLMs "work"—specifically stating that we don't offer a sufficient explanation of how LLMs go beyond "mansplaining as a service."

Their criticism seems to be motivated by disagreement with our position, rather than substantive engagement with our argument. They express skepticism about the efficacy of LLMs in general, to a stark degree that is at odds with the mainstream of the field. Our paper offers a specific position about how LLMs can be used; they offered their criticism on the basis that they don't like LLMs, not the quality of our ideas.

Furthermore, this reviewer claimed that our paper lacks significance because it doesn’t offer a full array of concrete technical solutions. But this is a position paper. We maintain that it is appropriate for the primary contribution of position paper to be critique and discussion. Concrete technical solutions are important, but are ultimately the purview of further work.

With that caveat in mind, we can make several modifications to address potential concerns of like-minded readers. Specifically, we plan to (1) cite papers that offer more direct examples of LLMs performing thick description-like tasks [1,2,3] and (2) more explicitly draw out assumptions in our argument about how thick descriptions might be aggregated.

**3-1 Review Response1:**

5qWL

**3-2 Reaction To Review1:**

This reviewer offered thoughtful, supportive commentary based on a nuanced analysis of our position. They considered multiple dimensions of our argument. This reviewer made a strong, good faith effort to articulate ways the paper could be improved, and our final paper will be improved as a result of their feedback. They exhibited no gatekeeping.

Thoughtfulness: 9/10
Supportive: 10/10
Technical vs Position: 9/10
Gate keeping: 10/10 (fully inclusive)

**3-3 Review Response2:**

fneG

**3-4 Reaction To Review2:**

This reviewer offered thoughtful, supportive commentary. They focused mainly on our position, but some of their stronger critiques seemed to be more about technical implementational details. However, they made a strong, good faith effort to articulate ways the paper could be improved. This reviewer considered multiple dimensions of our argument, and our final paper will be improved as a result of their feedback. They exhibited no gatekeeping.

Thoughtfulness: 9/10
Supportive: 9/10
Technical vs Position: 7/10
Gate keeping: 10/10

**3-5 Review Response3:**

hZRd

**3-6 Reaction To Review3:**

The commentary submitted by this reviewer was unthoughtful, unsupportive, and overall of poor quality. Their feedback was ultimately offered to convey the reviewer's idiosyncratic personal views, not in an effort to refine the final paper.

Generally, this reviewer brought a high degree of prejudice to their commentary, basing their response mainly on disagreement with our position rather than the strength of our argument. They focused almost exclusively on technical details rather than the substance of our position. Their critique was rooted in a view that LLMs primarily offer "mansplaining as a service" and faulted us for insufficiently providing a counterargument to this perspective.

We had three specific problems with their review:
(1) They fixated on a broader disagreement in the field about whether LLMs "work" in a general sense. Our position paper was not about adjudicating this very broad issue, on which the reviewer takes a position that deviates sharply from mainstream opinion.
(2) They didn't read our paper closely, seeming instead to have come to an early judgement based on a priori disagreement. Evidence: they summarized our paper as arguing "LLMs can be used to make thick descriptions legible at scale." Notably, this is the same phrasing as our subtitle, with "thick description" substituted (accurately but trivially) for "cultural context." They showed little engagement with substance in our paper beyond recapitulating a few keywords.
(3) The reviewer focused primarily on technical disagreements about what LLMs can do. They write that we treat LLMs as "magic, not a tool to be used to solve a problem raised." This is not true; our treatment of LLMs is constrained & empirically informed. Rather, this is probably their view on LLMs in general—a critique they would give to any pro-LLM paper.

Thoughtfulness: 2/10
Supportive: 2/10
Technical vs Position: 4/10
Gate keeping: 10/10

Note: we do not accuse this reviewer of violating the code of conduct.

---

### Meta-Review · Area_Chair_ujVs · 2025-09-13

**Rating:** 8
**Confidence:** 3

**Strengths:**

Summary:

The paper highlights the limitations of current LLMs with regard to understanding of cultural context and human meaning, primarily due to thin representations (in the form of numerical values); however, scholars in social sciences have already developed non-scalable frameworks for representing human meaning via thick representations/descriptions (verbal representations that capture heterogeneity and cultural context). Authors propose to combine the two approaches taking advantages of each. The paper outlines the key challenges in doing so and finally proposes method for developing better system can capture human experiences across domains.

Strengths:

1. The paper introduces new concepts from social sciences (including anthropology and political science) that will be beneficial for the AI community.
2. The paper is very well written.
3. The paper proposes alternative perspective well and in detail.

**Weaknesses:**

1. The paper does not satisfactorily address the legibility of meaning with regard to information aggregation.
2. Authors do not describe how and why LLMs can be appropriate tool to aggregate information for thick description.
3. The paper is lacking any empirical study

**Questions:**

1. Authors should explain in detail legibility of meaning as it is not clear from current formulation.
2. It is not clear how would the proposed system would be implemented in practice?

**Ethics:**

No major ethical concerns reported

**Thoroughness:**

1

---

### Decision · Program_Chairs · 2025-09-26

Reject